# Apparent Diffusion Coefficient as a Predictor of Microwave Ablation Response in Thyroid Nodules: A Prospective Study

**DOI:** 10.3390/diagnostics15192538

**Published:** 2025-10-09

**Authors:** Mustafa Demir, Yunus Yasar

**Affiliations:** Department of Radiology, Umraniye Training and Research Hospital, University of Health Sciences, Istanbul 34764, Turkey

**Keywords:** diffusion-weighted MR image, apparent diffusion coefficient, microwave ablation, benign thyroid nodules, volume reduction ratio

## Abstract

**Background:** Microwave ablation (MWA) is an effective, minimally invasive therapy for benign thyroid nodules; however, the treatment response varies considerably. Identifying imaging biomarkers that can predict volumetric outcomes may optimize patient selection. Diffusion-weighted MRI (DW-MRI) offers a noninvasive assessment of tissue microstructure through apparent diffusion coefficient (ADC) measurements, which may correlate with ablation efficacy. **Methods:** In this prospective study, 48 patients with 50 cytologically confirmed benign thyroid nodules underwent diffusion-weighted magnetic resonance imaging (DW-MRI) before minimally invasive ablation (MWA). Baseline ADC values were measured, and nodule volumes were assessed by ultrasound at baseline and 1, 3, and 6 months postprocedure. The volume reduction ratio (VRR) was calculated, and associations with baseline variables were analyzed via Pearson correlation and multivariable linear regression. ROC curve analysis was used to evaluate the diagnostic performance of ADC in predicting significant volume reduction (VRR ≥ 50%). **Results:** Lower baseline ADC values were strongly correlated with greater VRR at 3 months (r = −0.525, *p* < 0.001) and 6 months (r = −0.564, *p* < 0.001). Multivariable regression revealed that the baseline ADC was the sole independent predictor of the 6-month VRR (β = −19.52, *p* = 0.0004). ROC analysis demonstrated excellent discriminative performance (AUC = 0.915; 95% CI: 0.847–0.971), with an ADC cutoff of 2.20 × 10^−3^ mm^2^/s yielding 90.9% sensitivity and 83.3% specificity for predicting a favorable volumetric response. **Conclusions:** Baseline ADC values derived from DW-MRI strongly predict volumetric response following microwave ablation of benign thyroid nodules. Incorporating ADC assessment into preprocedural evaluation may enhance patient selection and improve therapeutic outcomes.

## 1. Introduction

Thyroid nodules are frequently observed in clinical practice, with ultrasonographic prevalence rates ranging from 19% to 67% in the general population [1,2]. Although most nodules are benign and asymptomatic, a subset necessitates intervention due to compressive symptoms, cosmetic concerns, or malignancy-related anxiety [3,4]. Surgery has traditionally served as the definitive treatment; however, associated risks—including recurrent laryngeal nerve injury, hypothyroidism, and visible scarring—as well as postoperative patient regret reported in some series, have driven increasing interest in minimally invasive alternatives [5,6,7,8].

Microwave ablation (MWA) has emerged as a safe and effective nonsurgical approach, offering rapid volume reduction, symptomatic relief, and high patient satisfaction [6,7,9]. Despite its clinical benefits, considerable variability in treatment response persists, with some nodules exhibiting suboptimal volume reduction. Accurate preprocedural prediction of therapeutic outcomes remains an unmet clinical need, which is critical for optimizing patient selection, informing clinical decision-making, and enhancing treatment success [5,7,10].

Diffusion-weighted magnetic resonance imaging (DW-MRI) enables noninvasive characterization of tissue microstructure by quantifying water diffusivity, represented by apparent diffusion coefficient (ADC) values [11,12]. While prior studies have primarily utilized DW-MRI to differentiate benign from malignant thyroid nodules [13,14], its role in predicting therapeutic response following thermal ablation remains largely unexplored.

Lower ADC values are associated with greater cellular density and restricted extracellular space [11,15], which are tissue features that may facilitate more efficient thermal energy absorption and greater volume reduction during the ablation process. Therefore, we hypothesized that baseline ADC measurements could be predictive biomarkers for the volumetric response following MWA.

This prospective study evaluated whether baseline ADC values derived from DW-MRI can independently predict treatment response, as measured by the volume reduction ratio (VRR), after MWA in benign thyroid nodules. By addressing a critical gap in preprocedural risk stratification, our findings may offer a new imaging biomarker to optimize clinical outcomes in thyroid nodule management.

## 2. Materials and Methods

### 2.1. Patient Selection and Study Design

This prospective study was approved by the local institutional ethics committee, and written informed consent was obtained from all participating patients. The study was conducted in accordance with the ethical principles outlined in the Declaration of Helsinki.

Patients with symptomatic benign thyroid nodules referred for MWA were prospectively enrolled between January 2023 and March 2024. All patients underwent comprehensive clinical assessment, ultrasonography, and ultrasound-guided fine-needle aspiration biopsy (FNAB) for cytological confirmation. All nodules were cytologically confirmed as Bethesda Category II (benign) on two separate FNABs. Ultrasound findings were also taken into consideration, and nodules with suspicious features or high TIRADS classification (≥TIRADS 4) were excluded from the study. The inclusion criteria were as follows: (a) age ≥18 years, (b) cytological confirmation of benignity via two FNABs and (c) refusal of surgery or surgical contraindication. The exclusion criteria included nodules with suspicious ultrasound features, prior thyroid surgery or ablation, pregnancy, or MRI contraindications.

The sample size was based on clinical feasibility rather than a priori power analysis. Demographic data, including age and sex, as well as baseline clinical characteristics, were collected. DW-MRI was performed within one week before ablation. Nodule volume was assessed via ultrasound at baseline and at 1, 3, and 6 months after the procedure. The volume was calculated via the ellipsoid formula:Volume = π/6 × length × width × depth.

### 2.2. Diffusion-Weighted MRI Protocol and Analysis

All DW-MRI examinations were conducted via a 1.5 Tesla MRI scanner (Siemens Healthineers, Erlangen, Germany) with a dedicated neck coil. An echo-planar imaging (EPI) sequence was used with the following parameters: repetition time (TR), 3500–4500 ms; echo time (TE), 60–90 ms; field of view (FOV), 200–240 mm; matrix size, 128 × 128; slice thickness, 3–4 mm; interslice gap, 0.5–1 mm; and number of excitations (NEX), 3–4 mm. Diffusion-sensitizing gradients were applied with a b-value of 800 s/mm^2^ in three orthogonal directions. The system automatically generated ADC maps.

The choice of b-value in DW-MRI directly affects the sensitivity and specificity of ADC measurements. Based on previous studies, b-values between 600 and 1000 s/mm^2^ provide an optimal balance between signal-to-noise ratio and diffusion sensitivity in thyroid imaging. Very high b-values (>1000 s/mm^2^) may lead to signal degradation, whereas low b-values (<400 s/mm^2^) may fail to detect subtle differences in diffusion. Therefore, a b-value of 800 s/mm^2^ was selected in this study to ensure both diagnostic accuracy and image quality, which is consistent with prior literature [11,13].

Circular regions of interest (ROIs), each measuring approximately 20–30 mm^2^, were carefully placed in the most considerable solid portion of the nodule on apparent diffusion coefficient (ADC) maps, meticulously avoiding areas of hemorrhage, necrosis, cystic degeneration, prominent vascular structures, and artifacts [11,13]. Measurements were taken on three separate slices of each nodule, and the mean ADC value was used for statistical analysis. All measurements were performed by a radiologist with over 5 years of experience in thyroid imaging, who was blinded to the clinical and ultrasound findings (Figure 1).

### 2.3. Microwave Ablation Procedure

Microwave ablation (MWA) was performed percutaneously under ultrasound guidance and local anesthesia. In cases where the nodule had a cystic or predominantly cystic component, complete aspiration of cystic fluid was performed before ablation. Aspiration was conducted under sterile conditions via an 18–20 gauge needle inserted into the cystic area under ultrasound guidance. A repeat ultrasound examination was performed to confirm fluid removal and to finalize preprocedural planning.

To minimize the risk of thermal injury to adjacent critical structures such as the carotid artery, jugular vein, and recurrent laryngeal nerve, hydrodissection was performed during the procedure. Under ultrasound guidance and sterile techniques, approximately 20–30 mL of 5% dextrose solution or normal saline was injected via a 20–22 gauge spinal or Chiba needle between the nodule and neighboring vital structures [3,4]. Adequate fluid spread around the nodule and separation of critical structures from the targeted area were confirmed ultrasonographically.

MWA was conducted via a CANYON microwave generator and a 16-gauge SURETIP antenna with a 3 mm active tip length (CANYON Medical, Nanjing, China). The antenna was carefully inserted into the center of the nodule under continuous ultrasound monitoring. The initial power output was set at 25 watts and gradually increased to 35, depending on intraprocedural findings such as increased echogenicity and microbubble formation. The ablation procedure was monitored in real-time via ultrasound and terminated when the transient hyperechoic cloud, caused by gas formation, encompassed the entire nodule, indicating sufficient thermal coverage [16,17].

### 2.4. Statistical Analysis

Continuous variables are expressed as the means ± standard deviations (SDs), and categorical variables are expressed as counts and percentages. The associations between continuous variables and the volume reduction ratio (VRR) were evaluated via Pearson correlation analysis. Differences in VRR between categorical groups were assessed via independent samples *t*-tests.

Initially, univariable analyses were performed to explore potential associations between baseline variables and VRR. Pearson correlation coefficients were calculated for continuous variables, whereas group comparisons were conducted for categorical variables. Variables with a *p*-value < 0.1 in the univariable analysis and clinically relevant factors previously reported in the literature to influence VRR were subsequently included in multivariable linear regression models to identify independent predictors of the 6-month VRR [11]. The diagnostic performance of baseline apparent diffusion coefficient (ADC) values in predicting significant volume reduction (defined as VRR ≥ 50% at 6 months) was assessed using receiver operating characteristic (ROC) curve analysis. The optimal ADC cutoff value was determined based on the Youden index.

All the statistical analyses were conducted via SPSS software (version 25.0; IBM Corp., Armonk, NY, USA), and a two-sided *p*-value < 0.05 was considered indicative of statistical significance.

## 3. Results

### 3.1. Baseline Characteristics

This prospective study included 48 patients with 50 benign thyroid nodules. The mean age of the patients was 45.6 ± 8.3 years, and the majority were female (58.0%). Nodules were most frequently located in the right lobe (52.0%), followed by the left lobe (36.0%) and the isthmus (12.0%).

In terms of composition, 64.0% of the nodules were solid, 24.0% were predominantly solid, and 12.0% were predominantly cystic. Intralesional vascularity was present in 46.0% of the nodules, whereas 54.0% were avascular. The average preprocedural nodule volume was 19.2 ± 15.7 mL, with a mean ablation duration of 12.2 ± 5.1 min. The mean ADC value measured via preprocedural diffusion-weighted MRI was 1.58 ± 0.46 × 10^−3^ mm^2^/s (Table 1).

### 3.2. Volume Reduction at 1, 3, and 6 Months Follow-Up

Following microwave ablation, a progressive and significant reduction in nodule volume was observed during follow-up. Mean VRR values increased steadily from baseline to 6 months, and the proportion of nodules achieving ≥50% volume reduction also rose over time (Table 2, Figure 2).

### 3.3. Predictors of Nodule Volume Reduction After Ablation

To evaluate the clinical and imaging variables potentially associated with treatment efficacy, the volume reduction ratio (VRR) at both the 3- and 6-month follow-ups was used as the primary outcome measure. The VRR was analyzed as a continuous variable, and its associations with constant and categorical baseline parameters were explored.

Among continuous variables, only the ADC demonstrated a significant negative correlation with VRR at both follow-up intervals, indicating that lower ADC values predicted greater volume reduction. Other parameters, including age, baseline nodule volume, and ablation duration, showed no significant associations (Table 3, Figure 3).

Analysis of categorical variables revealed no significant differences in VRR according to patient sex, vascularity, or nodule composition at either follow-up. Minor numerical variations between subgroups were not statistically significant (Table 4).

These findings suggest that preprocedural ADC values are the only significant predictor of nodule volume reduction after microwave ablation among various clinical and imaging features. Other baseline characteristics, including the structural and vascular features of the nodules, appear not to significantly influence the therapeutic response, as measured by the VRR at 3 and 6 months.

### 3.4. Multivariable Linear Regression Analysis

A multivariable linear regression analysis was performed using the 6-month volume reduction ratio (VRR) as the dependent variable to evaluate the independent influence of baseline parameters on ablation efficacy. Four predictors were included in the model: the mean ADC, preprocedural nodule volume, ablation duration, and presence of vascularity.

Among these variables, only the mean ADC value was significantly associated with the VRR. Specifically, the ADC exhibited moderate and adverse effects on treatment response (β = −19.52, 95% CI: −29.86 to −9.19, *p* = 0.0004), indicating that lower ADC values were independently associated with more significant volumetric reduction. In contrast, preprocedural nodule volume (β = +0.10, *p* = 0.75), ablation time (β = −0.71, *p* = 0.46), and the presence of vascularity (β = +1.92, *p* = 0.68) did not significantly contribute to the model (Table 5).

The results confirm that, when controlling for other clinically relevant parameters, the baseline ADC is the only independent predictor of the extent of volume reduction following microwave ablation. These findings provide quantitative support for the use of preprocedural diffusion-weighted MRI in predicting therapeutic efficacy.

### 3.5. Diagnostic Performance of ADC in Predicting the 6-Month Response

A receiver operating characteristic (ROC) curve analysis was conducted to assess the ability of the baseline apparent diffusion coefficient (ADC) values to predict the volumetric response to microwave ablation. A successful treatment outcome was a volume reduction ratio (VRR) of ≥50% at the 6-month follow-up. The analysis demonstrated excellent diagnostic performance, with an area under the curve (AUC) of 0.915 (95% CI: 0.847–0.971), reflecting a strong ability to distinguish between responders and non-responders (Figure 4).

The Youden index identified an ADC threshold of 2.20 × 10^−3^ mm^2^/s as the optimal cutoff point. When this threshold was applied, the sensitivity and specificity for predicting VRR ≥ 50% were 90.9% and 83.3%, respectively. These values indicate that the threshold correctly classified the majority of high and low responders in the cohort. The distributions of true positives, false positives, true negatives, and false negatives relative to this cutoff are consistent with a strong predictive model.

This cutoff was applied uniformly to all patients in the cohort, and its performance remained stable across the sample without the need for subgroup adjustments. These findings provide a clear and quantifiable threshold by which the baseline ADC can be used to evaluate postablation outcomes, independent of additional clinical parameters.

## 4. Discussion

Microwave ablation (MWA) has emerged as a well-tolerated and effective nonsurgical treatment for benign thyroid nodules, offering substantial volume reduction, symptom relief, and cosmetic improvement with minimal morbidity [18,19]. Despite these advantages, the variability in individual treatment response remains a clinical challenge. Identifying reliable preprocedural predictors of ablation success is essential for optimizing patient selection and improving therapeutic outcomes.

In this prospective study, we demonstrated that baseline ADC values derived from DW-MRI are strong independent predictors of volumetric response following MWA. Nodules with lower ADC values exhibited significantly greater volume reductions at the 3- and 6-month follow-ups. Multivariable regression confirmed the ADC as the sole independent predictor, and ROC analysis revealed excellent diagnostic performance (AUC = 0.915) with an optimal threshold of 2.20 × 10^−3^ mm^2^/s, indicating high sensitivity and specificity.

Our findings highlight the potential of DW-MRI not only as a diagnostic tool to distinguish benign from malignant thyroid lesions, but also as a predictive imaging biomarker for treatment planning. This application remains largely unexplored in the literature. Previous studies utilizing DW-MRI have focused primarily on characterizing thyroid nodules based on malignancy risk [19,20], whereas the ability of ADC values to predict therapeutic response has not been systematically investigated. To our knowledge, this is among the first prospective evaluations proposing the ADC as a predictor of MWA outcomes in benign thyroid nodules.

The biological rationale supporting our observations is grounded in the microstructural characteristics captured by ADC measurements. Lower ADC values reflect greater cellularity and restricted extracellular space [14,15,19,21], which are tissue properties that may enhance thermal energy absorption and facilitate more effective ablation-induced necrosis [22]. Our results corroborate this hypothesis, suggesting that nodules with denser cellular architecture respond more favorably to thermal therapy.

Among various baseline clinical and imaging parameters, only the ADC was significantly associated with volume reduction, including patient age, nodule volume, vascularity, composition, and ablation duration. These findings emphasize the limitations of conventional ultrasound features in predicting therapeutic response and suggest that microstructural tissue profiling via DW-MRI may offer superior prognostic information. Previous reports have identified various factors, such as nodule size or vascularity, that influence ablation outcomes, but these associations were not observed in our cohort [18,23]. Zhi et al. (2018) [23] suggested that smaller nodules exhibit more significant volume reduction following microwave ablation. Luo et al. (2021) [18] identified nodule composition—specifically the proportion of solid components—as an essential determinant of treatment response. However, in our cohort, neither initial nodule volume nor initial nodule composition demonstrated a significant association with postablation volume reduction. These discrepancies may stem from differences in imaging techniques, nodule selection criteria, or sample sizes, underscoring the need for standardized protocols in future investigations.

Recent studies have further investigated the predictive value of ultrasound-derived parameters in estimating the volumetric response of benign thyroid nodules following ablation. Baseline nodule volume, composition, echogenicity, and vascularity have been reported as significant determinants of post-procedural volume reduction in some cohorts. Meanwhile, emerging US-based radiomics and deep learning models have also demonstrated strong performance in forecasting suboptimal volume reduction outcomes (<50%) [24,25,26]. Although these ultrasound-based approaches are more accessible and cost-effective, they primarily reflect macroscopic morphology. In contrast, ADC values obtained from DW-MRI provide microstructural information about tissue diffusion, potentially capturing subtle biological variations that influence the efficiency of ablation. In this regard, DW-MRI may complement ultrasound by providing an additional, quantifiable imaging biomarker for predicting preprocedural response.

ROC analysis further reinforced the clinical utility of the ADC measurement. The identified cutoff of 2.20 × 10^−3^ mm^2^/s provided robust discrimination between responders and non-responders, offering a practical threshold for clinical decision-making. The incorporation of DW-MRI into preablation assessments could assist in identifying ideal candidates for MWA, tailoring treatment strategies, and improving overall therapeutic efficiency.

In addition to its ability to predict ablation response, DW-MRI may provide added value in the preprocedural diagnostic evaluation of thyroid nodules. Fine-needle aspiration biopsy (FNAB) remains the standard approach for confirming benignity before ablation; however, its diagnostic accuracy can be limited, particularly for large or heterogeneous nodules, where sampling may not capture the complete histological profile [22,27,28]. Although our findings suggest that DW-MRI may offer complementary information by revealing diffusion characteristics beyond those detected by FNAB, this interpretation remains speculative, as all nodules included in our study were cytologically benign. Therefore, further research involving histologically confirmed malignant cases is required to validate this potential diagnostic role. This cautious interpretation highlights the need for larger, prospective studies to investigate the potential dual role of DW-MRI in both treatment planning and pre-ablation risk assessment.

Several limitations of this study warrant consideration. First, the sample size was relatively small and was derived from a single center, which may affect the generalizability of the findings. Second, the follow-up period was limited to six months, which precluded the assessment of longer-term durability of response. Third, DW-MRI was performed using a single b-value (800 s/mm^2^), thereby restricting the exploration of more advanced diffusion models. Finally, the ADC measurements were based on manual region-of-interest placement, which introduced potential interobserver variability; however, care was taken to standardize the measurement techniques.

## 5. Conclusions

In conclusion, our prospective study demonstrated that baseline ADC values derived from DW-MRI are strong, independent predictors of the volumetric response following microwave ablation of benign thyroid nodules. Lower ADC values consistently predicted superior volume reduction. These findings suggest that integrating DW-MRI into preprocedural evaluation protocols may enhance patient selection and optimize therapeutic outcomes. Larger multicenter studies with more extended follow-up periods are warranted to validate these results and establish standardized imaging protocols for clinical application.

## Figures and Tables

**Figure 1 diagnostics-15-02538-f001:**
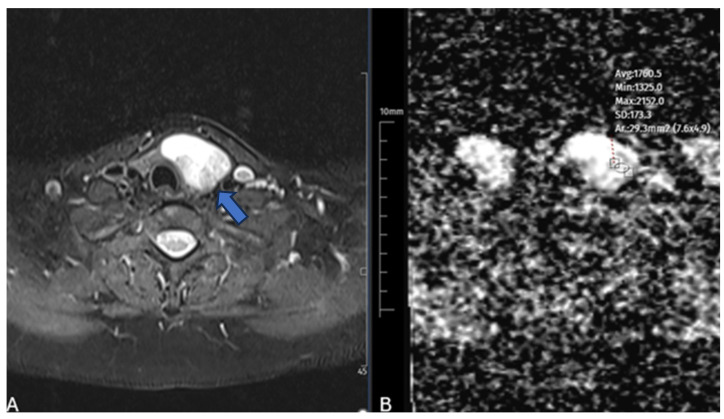
A 47-year-old female patient with a benign nodule. A, B T2-weighted MR image (**A**) demonstrates a hyperintense benign thyroid nodule (arrow). Corresponding axial ADC map (**B**) shows placement of a region of interest (ROI) for apparent diffusion coefficient (ADC) measurement in the solid portion of the nodule, with ADC value measured as 1.76 × 10^−3^ mm^2^/s.

**Figure 2 diagnostics-15-02538-f002:**
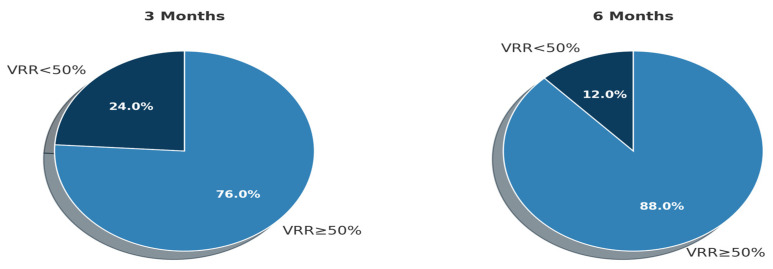
Distribution of nodules achieving significant volume reduction (VRR ≥ 50%) at the 3-month follow-up after microwave ablation.

**Figure 3 diagnostics-15-02538-f003:**
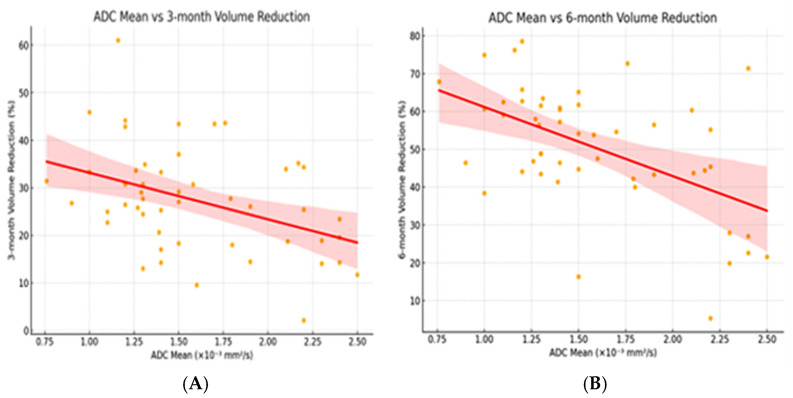
Scatter plots demonstrating the correlations between ADC values and volume reduction ratios at 3 months (**A**) and 6 months (**B**). Both correlations were moderately negative and statistically significant (3 months: Pearson’s r = −0.52, *p* < 0.001; 6 months: Pearson’s r = −0.56, *p* < 0.001). Orange dots represent individual data points, and the red line indicates the linear regression fit.

**Figure 4 diagnostics-15-02538-f004:**
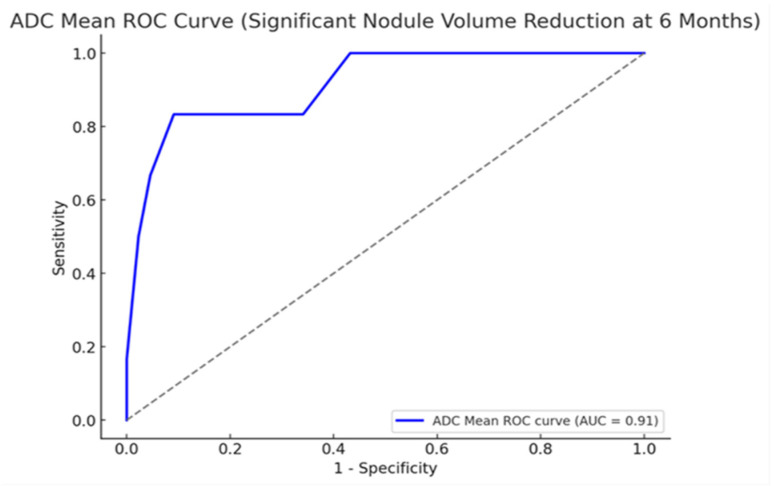
ROC curve illustrating the predictive performance of mean ADC values for significant nodule volume reduction (VRR ≥ 50%) at the 6-month follow-up (AUC = 0.91). The dashed diagonal line represents the reference line of no-discrimination (AUC = 0.5).

**Table 1 diagnostics-15-02538-t001:** Demographic and Nodule Characteristics of Patients.

Variable	Value
No. of patients	48
Mean age (years)	45.6 ± 8.3
Sex (F/M)	28/20 (58.0%/42.0%)
No. of nodules	50
Mean nodule volume (mL)	19.2 ± 15.7
Ablation duration (minutes)	12.2 ± 5.1
ADC mean ± SD (×10^−3^ mm^2^/s)	1.58 ± 0.46
**Growth location**	
Left lobe	18 (36.0%)
Right lobe	26 (52.0%)
Isthmus	6 (12.0%)
**Nodule component**	
Solid	32 (64.0%)
Predominantly solid	12 (24.0%)
Predominantly cystic	6 (12.0%)
**Vascularization**	
Present	23 (46.0%)
Absent	27 (54.0%)

Note: ADC, Apperant Diffusion Coefficient; SD, Standard Deviation.

**Table 2 diagnostics-15-02538-t002:** Thyroid nodule volumes and volume reduction rates (VRRs) at baseline and during follow-up after microwave ablation.

Time	Mean Volume ± SD (mL)	Mean VRR (%) ± SD
Baseline	19.16 ± 15.67	–
1 Month	9.82 ± 9.15	48.16 ± 19.13
3 Month	7.17 ± 7.19	61.64 ± 17.38
6 Month	5.28 ± 6.36	73.11 ± 17.40

Note: VRR, Volume Reduction Rate; SD, Standard Deviation; mL, Milliliter.

**Table 3 diagnostics-15-02538-t003:** Correlation Between Continuous Baseline Variables and Nodule Volume Reduction.

Variable	Pearson r—3rd Month	Pearson r—6th Months
ADC Mean	−0.525 (*p* < 0.001) *	−0.564 (*p* < 0.001) *
Age	−0.159 (*p* = 0.270) *	−0.003 (*p* = 0.980) *
Nodule Volume	0.066 (*p* = 0.650) *	−0.049 (*p* = 0.730) *
Ablation Duration	−0.034 (*p* = 0.810) *	−0.168 (*p* = 0.240) *

Note: ADC, Apparent Diffusion Coefficient, * Pearson Correlation Analysis.

**Table 4 diagnostics-15-02538-t004:** Groupwise Comparison of Categorical Variables by Mean Volume Reduction.

Variable	Mean VRR (3 Months)	Mean VRR (6 Months)
Gender (F/M)	64.8%/57.4% (*p* = 0.116) *	76.0%/69.1% (*p* = 0.275) *
Vascularity (Yes/No)	63.7%/60.0% (*p* = 0.465) *	77.9%/69.0% (*p* = 0.108) *

* *t*-tests.

**Table 5 diagnostics-15-02538-t005:** Multivariable Linear Regression Analysis of Factors Associated with 6-Month Volume Reduction Ratio (VRR).

Variable	β Coefficient	Standard Error	*p*-Value	95% CI (Lower–Upper)
Constant	109.83	10.74	<0.001	88.78 to 130.87
ADC Mean	−19.52	5.13	0.0004	−29.58 to −9.46
Nodule Volume	0.10	0.31	0.749	−0.51 to 0.71
Ablation Duration	−0.71	0.94	0.455	−2.56 to 1.14
Vascularity	1.92	4.58	0.677	−7.06 to 10.89

## Data Availability

The data presented in this study are available from the corresponding author due to ethical restrictions.

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
