# Peer review of "Apparent Diffusion Coefficient as a Predictor of Microwave Ablation Response in Thyroid Nodules: A Prospective Study"

_diagnostics, 2025, doi:10.3390/diagnostics15192538_

Round 1
Reviewer 1 Report
Comments and Suggestions for Authors
The article “Apparent Diffusion Coefffcient as a Predictor of Microwave Ablation Response in Thyroid Nodules: A Prospective Study”evaluated whether baseline ADC values derived from DW- MRI can independently predict treatment response, as measured by the volume reduction ratio (VRR), after MWA in benign thyroid nodules. With the high prevalence of diseases related to thyroid nodules, research in this area is of great clinical relevance. Here are a few comments for the authors’ consideration:
- “however, associated risks—including recurrent laryngeal nerve injury, hypothyroidism, and visible scarring—have driven increasing interest in minimally invasive alternatives [5-7].”The study (PMID: 40057484) also found that some patients experienced regret following thyroid surgery. Please cite it.
- “In this prospective study, 48 patients with 50 cytologically confirmed benign thyroid nodules underwent diffusion-weighted magnetic resonance imaging (DW-MRI) before minimally invasivethyroidectomy (MWA). ” “thyroidectomy” should be “” Correction required.
- In Table 1, what constitutes the remaining 6% in the nodule component? Are they purely cystic, or do they represent missing data? Clarification is needed to ensure data accuracy.
- The author presents an intriguing perspective that DW-MRI may complement the limitations of FNAB in detecting potentially malignant tumors. This represents a significant potential clinical value; however, since all nodules in this study were cytologically benign, this claim has not been directly validated. Greater caution should be exercised when advancing this argument, explicitly stating that it remains speculative and requires confirmation in future studies.
Author Response
Please see the attachment.
--
Dear Editor and Reviewers,
We sincerely thank you for your valuable time and constructive comments on our manuscript entitled “Apparent Diffusion Coefficient as a Predictor of Microwave Ablation Response in Thyroid Nodules: A Prospective Study.”
We carefully reviewed all suggestions and revised the manuscript accordingly to improve its clarity, accuracy, and scientific rigor. Each comment has been addressed point by point below, followed by a detailed response and description of the corresponding revisions.
All modifications made in the manuscript have been highlighted in track changes in the revised Word file for your convenience.
Response to Reviewer #1
Comment 1:
“However, associated risks—including recurrent laryngeal nerve injury, hypothyroidism, and visible scarring—have driven increasing interest in minimally invasive alternatives [5–7].” The study (PMID: 40057484) also found that some patients experienced regret following thyroid surgery. Please cite it.
Response:
The sentence in the Introduction has been revised to incorporate the reviewer’s suggested reference regarding postoperative regret following thyroid surgery.
Revision in manuscript: Corrected in the Introduction, lines 40-44, and in the References, lines 375-376:
“Surgery has traditionally served as the definitive treatment; however, associated risks—including recurrent laryngeal nerve injury, hypothyroidism, and visible scarring—as well as postoperative patient regret reported in some series, have driven increasing interest in minimally invasive alternatives (5–8).”
Comment 2:
“In this prospective study, 48 patients with 50 cytologically confirmed benign thyroid nodules underwent diffusion-weighted magnetic resonance imaging (DW-MRI) before minimally invasive thyroidectomy (MWA).” “thyroidectomy” should be “” Correction required.
Response:
We thank the reviewer for catching this typographical error. The word “thyroidectomy” has been corrected to “microwave ablation.”
Revision in manuscript: Corrected in the Abstract, line 18:
“... underwent diffusion-weighted magnetic resonance imaging (DW-MRI) before microwave ablation (MWA).”
Comment 3:
“In Table 1, what constitutes the remaining 6% in the nodule component? Are they purely cystic, or do they represent missing data? Clarification is needed to ensure data accuracy.”
Response:
We thank the reviewer for carefully noting this inconsistency. The discrepancy resulted from a minor data omission: three nodules were mistakenly not included in the component breakdown. Specifically, two of these nodules were solid, and one was predominantly cystic. The total number of nodules (N = 50) remains unchanged, and this correction does not affect any statistical analyses or conclusions. Table 1 and the related sentence in the Results section have been updated accordingly to ensure data accuracy and internal consistency.
Revision in manuscript: Corrected in Table 1, line 172, and in the Results, lines 166-167:
Table 1 (Nodule component) has been corrected as follows:
- Solid: 32 (64.0%)
- Predominantly solid: 12 (24.0%)
- Predominantly cystic: 6 (12.0%)
Results section (Baseline Characteristics):
“In terms of composition, 64.0% of the nodules were solid, 24.0% were predominantly solid, and 12.0% were predominantly cystic.”
Comment 4:
The author presents an intriguing perspective that DW-MRI may complement the limitations of FNAB in detecting potentially malignant tumors. This represents a significant potential clinical value; however, since all nodules in this study were cytologically benign, this claim has not been directly validated. Greater caution should be exercised when advancing this argument, explicitly stating that it remains speculative and requires confirmation in future studies.
Response:
We appreciate the reviewer’s insightful comment. We agree that our statement regarding the potential role of DW-MRI in detecting malignancy should be presented more cautiously, as our study exclusively included cytologically benign nodules. Accordingly, the relevant paragraph in the Discussion section has been revised to clearly acknowledge that this interpretation is speculative and requires further validation in future research involving histologically confirmed malignant cases.
Revision in manuscript: Corrected in the Discussion, lines 316-322:
“Although our results suggest that DW-MRI may provide complementary information to FNAB by revealing tissue diffusion characteristics, this interpretation remains speculative. As our study included only cytologically benign nodules, further research involving histologically confirmed malignant cases is necessary to validate this potential diagnostic role. This cautious interpretation highlights the need for larger, prospective studies to investigate the potential dual role of DW-MRI in both treatment planning and pre-ablation risk assessment.”
Reviewer 2 Report
Comments and Suggestions for Authors
The article clearly presents the thesis of the title "Apparent Diffusion Coefficient as a Predictor of Microwave Ablation Response in Thyroid Nodules: A Prospective Study." Conclusions are supported by presented results. The data analysis is correct. The number of patients is sufficient. The novelty is high.
However, I have some comments:
1. In the abstract there is a doubled explanation of the ADC abbreviation.
2. Which Bethesda categories confirmed benign nodules in FNA cytology? Was USG taken into consideration during enrollment of patients to the study? Did you exclude high TIRADS patients?
3. The text could be simplified by reducing some details presented further in the tables (e.g., lines 174-179, 194-199, and 230-234).
4. All presented figures are necessary, but Figs. 2 and 3 are blurred.
5. The discussion could be enriched by recent literature on ultrasonography-related values impact on VRR after MVA - the USG is more accessible. What can patients get with additional DW-MRI?
Author Response
Please see the attachment.
--
Dear Editor and Reviewers,
We sincerely thank you for your valuable time and constructive comments on our manuscript entitled “Apparent Diffusion Coefficient as a Predictor of Microwave Ablation Response in Thyroid Nodules: A Prospective Study.”
We carefully reviewed all suggestions and revised the manuscript accordingly to improve its clarity, accuracy, and scientific rigor. Each comment has been addressed point by point below, followed by a detailed response and description of the corresponding revisions.
All modifications made in the manuscript have been highlighted in track changes in the revised Word file for your convenience.
Response to Reviewer #2
Comment 1:
“In the abstract there is a doubled explanation of the ADC abbreviation.”
Response:
We thank the reviewer for noting this oversight. The abbreviation “ADC (apparent diffusion coefficient)” was indeed defined twice in the abstract—once in the title context and once again in the body of the text. The redundant explanation has now been removed for clarity and stylistic consistency.
Revision in manuscript: Corrected in the Abstract, Line 19:
“Baseline ADC values were measured, and nodule volumes were assessed by ultrasound at baseline and 1, 3, and 6 months postprocedure.”
Comment 2:
“Which Bethesda categories confirmed benign nodules in FNA cytology? Was USG taken into consideration during enrollment of patients to the study? Did you exclude high TIRADS patients?”
Response:
We appreciate the reviewer’s thoughtful questions regarding patient selection. All nodules included in this study were cytologically confirmed as Bethesda Category II (benign) on two separate fine-needle aspiration biopsies (FNABs). In addition, ultrasonographic features were considered during enrollment to exclude nodules with suspicious characteristics. Specifically, nodules demonstrating high-risk sonographic features or classified as high TIRADS (≥ TIRADS 4) were excluded. This information has now been clarified in the Materials and Methods section to improve transparency of the inclusion criteria.
Revision in manuscript: Corrected in the Materials and Methods, Lines 76-79:
“All nodules were cytologically confirmed as Bethesda Category II (benign) on two separate FNABs. Ultrasound findings were also taken into consideration, and nodules with suspicious features or high TIRADS classification (≥ TIRADS 4) were excluded from the study.”
Comment 3:
“The text could be simplified by reducing some details presented further in the tables (e.g., lines 174–179, 194–199, and 230–234).”
Response:
We thank the reviewer for this helpful suggestion. We agree that some portions of the Results section repeat numerical data already presented in the tables. To improve clarity and readability, these sections have been condensed to summarize only the key findings, while detailed numerical values are presented in Tables 2–4. This adjustment enhances the flow of the text and avoids redundancy without altering any results or conclusions.
Revision in manuscript:
Simplifications were made in the Results section as follows:
Lines 175–178:
Revised text:
“Following microwave ablation, a progressive and significant reduction in nodule volume was observed during follow-up. Mean VRR values increased steadily from baseline to 6 months, and the proportion of nodules achieving ≥50% volume reduction also rose over time (Table 2, Figure 2).”
Lines 191–194:
Revised text:
“Among continuous variables, only the ADC demonstrated a significant negative correlation with VRR at both follow-up intervals, indicating that lower ADC values predicted greater volume reduction. Other parameters, including age, baseline nodule volume, and ablation duration, showed no significant associations (Table 3, Figure 3).”
Lines 197–199:
Revised text:
“Analysis of categorical variables revealed no significant differences in VRR according to patient sex, vascularity, or nodule composition at either follow-up. Minor numerical variations between subgroups were not statistically significant (Table 4).”
Comment 4:
“All presented figures are necessary, but Figs. 2 and 3 are blurred.”
Response:
We thank the reviewer for noticing this issue. The low resolution of Figures 2 and 3 was due to compression during manuscript preparation. Both figures have now been replaced with high-resolution versions (300 dpi, TIFF format) to ensure optimal clarity in print and online publication. The figure captions and numbering remain unchanged.
Revision in manuscript:
- Figure 2 and Figure 3 have been replaced with high-resolution images (300 dpi, TIFF).
- No textual modifications were required.
Comment 5:
“The discussion could be enriched by recent literature on ultrasonography-related values' impact on VRR after MWA – the USG is more accessible. What can patients get with additional DW-MRI?”
Response:
We thank the reviewer for this excellent suggestion. The Discussion has been expanded to summarize recent studies showing that ultrasound-based parameters (baseline volume, composition, echogenicity, vascularity) and US-derived radiomics/AI models can predict VRR after thermal ablation. Nevertheless, in our cohort, pre-procedural ADC remained an independent predictor of VRR, suggesting that DW-MRI provides additional microstructural insight beyond US. This revision highlights the complementary roles of US and DW-MRI.
Revision in manuscript: Corrected in the Discussion section, Lines 293-304, and in the References section, Lines 417-423:
“Recent studies have further explored the predictive role of ultrasound-derived parameters in estimating the volumetric response of benign thyroid nodules after ablation. Baseline nodule volume, composition, echogenicity, and vascularity have been reported as significant determinants of post-procedural volume reduction in some cohorts. Meanwhile, emerging US-based radiomics and deep learning models have also demonstrated strong performance in forecasting suboptimal volume reduction outcomes (<50%) [24-26]. Although these ultrasound-based approaches are more accessible and cost-effective, they primarily reflect macroscopic morphology. In contrast, ADC values obtained from DW-MRI provide microstructural information about tissue diffusion, potentially capturing subtle biological variations that influence the efficiency of ablation. In this regard, DW-MRI may complement ultrasound by providing an additional, quantifiable imaging biomarker for predicting preprocedural response.”